# FaceParts: Segmentation and Editing of Gaussian Splatting Avatars*

Figure 1: Overview of the FaceParts reconstruction pipeline. The unsupervised segmentation process has three stages: **Disentanglement**, where a neural network groups similar Gaussians into shared facial components; **Refinement**, where density-based clustering extracts coherent subparts; and **Extraction**, where these parts are stored for reuse. The framework supports **facial part swapping**, enabling modification of a target avatar by transferring features (e.g., beard, eyes).

## Abstract

Facial editing is an important task with applications in entertainment, virtual reality, and digital avatars. Most existing approaches rely on generative models in the 2D image domain, while 3D the task is typically performed through labor-intensive manual editing. We propose FaceParts, a framework for *unsupervised segmentation and editing of Gaussian Splatting avatars*. Unlike existing 2D or mesh-assisted methods, our approach operates directly in the Gaussian domain, decomposing avatars into semantically coherent facial parts without supervision. The method integrates feature disentanglement, density-based clustering, and FLAME-anchored part transfer, enabling precise editing and cross-avatar part swapping. Experiments on the NeRSemble dataset with 11 subjects demonstrate robust isolation of features such as beards, eyebrows, eyes, and mustaches. Quantitative evaluation confirms that transferred segments adapt to pose and expression, while maintaining identity consistency (ID = 0.943), low Average Expression Distance (AED = 0.021), and low Average Pose Distance (APD = 0.004).

---

*Codebase available at github.com/FaceSwap-ICLR

## 1    INTRODUCTION

The growing interest in virtual and augmented reality has significantly advanced the need for realistic and manipulable 3D avatars, particularly in fields such as gaming, video calls, and personalized biometry (Islam & Wang, 2025). The ability to generate and edit 3D avatars with high fidelity and precision plays a crucial role in enhancing user experience in various applications, from gaming to healthcare, where accurate digital representations can influence interaction quality and accessibility (Zhang et al., 2024). One of the key challenges in avatar creation and manipulation is achieving seamless and fine-grained facial editing, particularly with respect to features such as beard, mustache, and hair, which can be difficult to isolate with traditional methods Zhang et al. (2023).

Current techniques primarily focus on 2D representations, where facial features are often edited using generative models like GANs (Li et al., 2025) or manipulated within latent spaces (Suwała et al., 2024). However, these approaches often fail to deliver the precision required for realistic and immersive 3D avatar creation. Additionally, they suffer from entanglement of facial features and identity loss, which is usually mitigated by adding additional loss terms (Yao et al., 2021). In the case of 3D avatars, most methods rely on labor-intensive manual editing, which is not scalable for dynamic environments requiring real-time interaction. Moreover, existing segmentation methods are typically confined to continuous regions (Wang et al., 2024b; Xie et al., 2021), making it difficult to work with distinct facial features at a granular level. Adopting 2D segmentation models (Chen et al., 2025) does not guarantee the multi-view consistency that is crucial for 3D models.

In this paper, we present FaceParts novel framework to address these limitations by introducing unsupervised segmentation and localized editing for 3D avatars (see Figure 1). An additional neural network obtains unsupervised segmentation (see Figure 2), which maps the centers of 3D Gaussian splats to their parameters: color, opacity, and covariance matrix. Operating directly on a 3D representation avoids resource-intensive rendering and enables efficient learning with a compact network architecture. The final bottleneck layer employs Gumbel-Softmax (Herrmann et al., 2020), which forces the model to select only one channel per Gaussian component. As a result, our network produces natural segmentation. This segmentation allows for a possibility of geometrical manipulation by switching a set of 3D Gaussians between face avatars (see Figure 3).

FaceParts allows for the decomposition of 3D Gaussian Splatting avatars into semantically meaningful 3D segments, facilitating precise control over individual facial features while preserving global consistency and rendering quality. Furthermore, the framework allows for seamless editing, part swapping, and transfer of attributes such as facial hair, enabling avatar customization with applications ranging from personalized digital avatars in virtual environments to accurate biometric representations in healthcare.

In summary, our principal contributions are as follows.

- We introduce FaceParts, a new model for unsupervised face segmentation that leverages an MLP fused with a Hash Grid to produce interpretable segments,
- Our model enables natural manipulation of facial segments without compromising subject identity or unintentionally modifying other attributes,
- We provide a comprehensive evaluation protocol combining identity, pose, expression, and realism metrics to assess segment transfer.

## 2    RELATED WORK

**Gaussian Avatars.**    Gaussian Splatting is an efficient technique for 3D scene representation, outperforming volumetric and neural methods such as NeRF (Mildenhall et al., 2020). HumanGaussian Splatting (Moreau et al., 2024) provides an end-to-end framework for training and animating human models. Since pure splat-based methods often overfit and introduce artifacts, hybrid approaches are used. SplattingAvatar (Shao et al., 2024) combines meshes for motion with 3D Gaussians for fine detail, enabling diverse animations. FlashAvatar (Xiang et al., 2024) accelerates training and rendering through UV-space parameterization and improved initialization. Gaussian Avatars (Qian et al., 2024) focus on head models, achieving controllable expression, pose, and viewpoint by assigning and optimizing 3D Gaussians on mesh triangles.

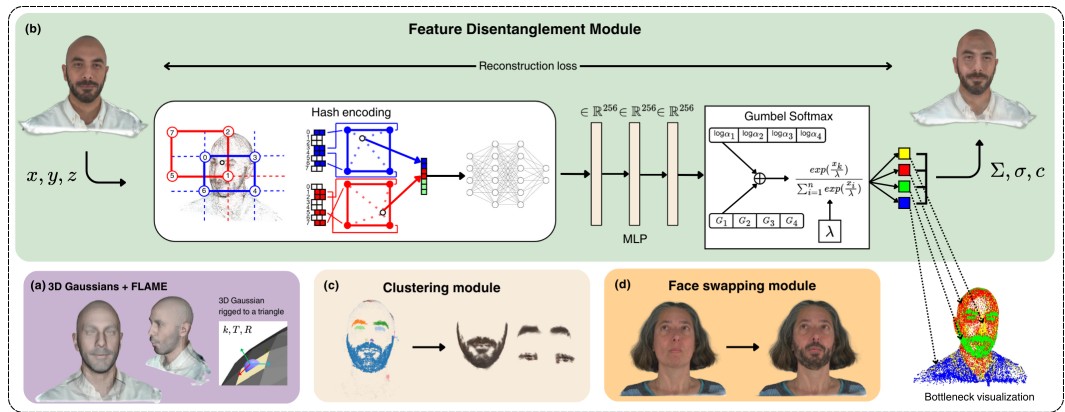

Figure 2: Framework overview: (a) Head avatars employ a hybrid solution of 3D Gaussians rigged to a parametric mesh model. (b) Reconstructing Gaussian features based solely on position encourages grouping of similar semantic regions within the same bottleneck channel. (c) Density-based clustering ensures that the components constitute smaller logical facial parts, such as an eyebrow. (d) The underlying FLAME model enables transferring facial parts without manual adjustments.

**Gaussian Splatting for Editable Avatars.** Editable 3D head avatars are valuable for VR/AR, allowing intuitive modification of attributes such as eye color or hair. PERSE (Cha et al., 2025) generates 3D Gaussian avatars from a single portrait with disentangled, text-controllable attributes, but relies on CLIP and segmentation networks. EG³D (Li et al., 2025) splits Gaussian splats into facial and non-facial groups, integrating the FLAME mesh and texture maps for facial details while constraining geometry for regions like hair. While not face-dedicated, (Li et al., 2023) presented a high-fidelity neural surface reconstruction framework that combines multi-resolution hash grids with signed distance function rendering, enabling detailed and large-scale 3D mesh reconstruction from RGB images without auxiliary depth or segmentation data. Guédon & Lepetit (2024) extracted accurate, editable meshes directly from Gaussian Splatting representations by aligning Gaussians to surfaces, enabling fast mesh reconstruction, joint optimization, and realistic. Li et al. (2024b) introduce 3D Gaussian GANs for head avatars, while Sakamiya et al. (2024) present AvatarPerfect, a system for constructing and refining avatars. These methods enable identity-preserving reconstructions and manual edits but do not achieve automatic semantic face-part decomposition.

**Unsupervised Facial Segmentation.** Unsupervised face decomposition has been explored mainly in 2D. Yu et al. (2022) propose hierarchical parsing for face parts, and general unsupervised segmentation approaches (Niu et al., 2023; Yang et al., 2023) demonstrate the emergence of semantically consistent regions. Although effective for discovering components such as eyes, nose, or beard, these methods have not been extended to 3D Gaussian splatting avatars. Wang et al. (2024b) propose a 3D face reconstruction method guided by facial part segmentation through a novel Part Re-projection Distance Loss, which improves alignment of reconstructed facial components and enables accurate modeling of extreme expressions.

**Facial Attribute Editing and Part Swapping.** Facial attribute editing has been extensively studied in 2D image domain (He et al., 2019; Guo et al., 2021; Aliari et al., 2023; Zhang et al., 2022), enabling local control of features such as hair, mustaches, or accessories via GAN-based optimization or semantic manipulation. However, existing work is restricted to 2D or latent spaces. To our knowledge, our approach is the first to achieve *unsupervised face-part segmentation and swapping directly in the Gaussian Splatting domain*, supporting localized 3D facial editing across avatars.

## 3 PRELIMINARY

**Gaussian Splatting** Gaussian Splatting (GS) represents a 3D scene using a collection of Gaussian primitives in three-dimensional space. Formally, the scene is described as a set of components

$$\mathcal{G}_{GS} = \{(\mathcal{N}(\boldsymbol{\mu}_i, \boldsymbol{\Sigma}_i), \sigma_i, \mathbf{c}_i)\}_{i=1}^{n},$$

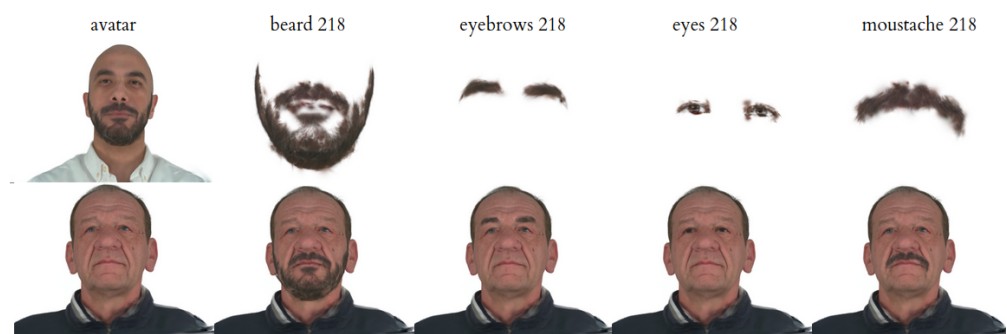

Figure 3: Examples of face-part transfer across identities. The top row shows extracted segments (beard, eyebrows, eyes, and mustache) from avatar 218, while the bottom row shows the same parts attached to avatar 210.

where $\mu_i$ is the centroid location, $\mathbf{\Sigma}_i$ is the covariance matrix describing its spatial extent, $\sigma_i$ denotes opacity, and $\mathbf{c}_i$ contains the spherical harmonics (SH) color coefficients of the $i$-th Gaussian. (Kerbl et al., 2023). The high efficiency of GS arises from its rendering strategy, which splats (projects) the Gaussians directly onto the image plane. Training alternates between rendering with the current Gaussian set and minimizing the error with respect to the training views.

**Flame** Faces Learned with an Articulated Model and Expressions(FLAME) (Li et al., 2017) is a parametric 3D face model built from thousands of carefully registered facial scans. It factorizes shape into identity, pose, and expression, mirroring strategies used in full-body modeling. The representation is designed to be lightweight and compatible with standard rendering pipelines.

**Avatar representation** As the underlying avatar representation, we employ Gaussian Avatars (Qian et al., 2024), which embed 3D Gaussian Splatting into the FLAME model to produce realistic and animated avatars (see Fig 4) by associating each Gaussian splat with a mesh triangle and expressing its position, rotation, and scale in a local coordinate system. Such a formulation provides natural regularization of the splat parameters during the joint optimization of splats and mesh.

## 4 FACEPARTS

Our framework decomposes a Gaussian Splatting avatar into semantically meaningful facial parts and enables their transfer across subjects. The workflow consists of 3 main stages: (1) *avatar feature disentanglement*, where Gaussians are divided into groups based on similar appearance properties; (2) *group clustering*, which refines these groups into coherent and well-defined facial segments; and (3) *face swap*, where extracted parts such as a beard or mustache are transferred to a target avatar through FLAME parameterization. An overview of the process is shown in Figure 2.

**Avatar Feature Disentanglement** We operate directly on trained Gaussian avatars without requiring labels or external supervision. Each Gaussian is described by its 3D position and a 56-dimensional feature vector encoding rotation, scale, opacity, and spherical harmonics. In Gaussian Avatars the parameters of each Gaussian are stored in a triangle-local coordinate system of the underlying FLAME mesh, i.e., positions, scales, and rotations are defined relative to a parent triangle. Since our disentanglement model requires consistent global positions, we first convert local parameters to global ones. Following (Qian et al., 2024), the transformation is given by

$$s' = ks, \qquad \mu' = R\mu + T, \qquad r' = Rr, \tag{1}$$

where $s, \mu, r$ are the local scale, position, and rotation of a Gaussian, and $k, R, T$ denote the scale, orientation, and center of its parent triangle. Using global values anchors all variation within a single shared space, providing a stable and consistent basis for disentangling visual attributes. This helps the network to explain changes in scale, pose, and translation relative to one common representation, which improves the quality of representation (Chen et al., 2021).

NeRF (Mildenhall et al., 2020) shows that learning directly on the $xyz$ components performs poorly on network $f_\theta$ and it uses encoding function to map inputs to higher dimensional space. For this

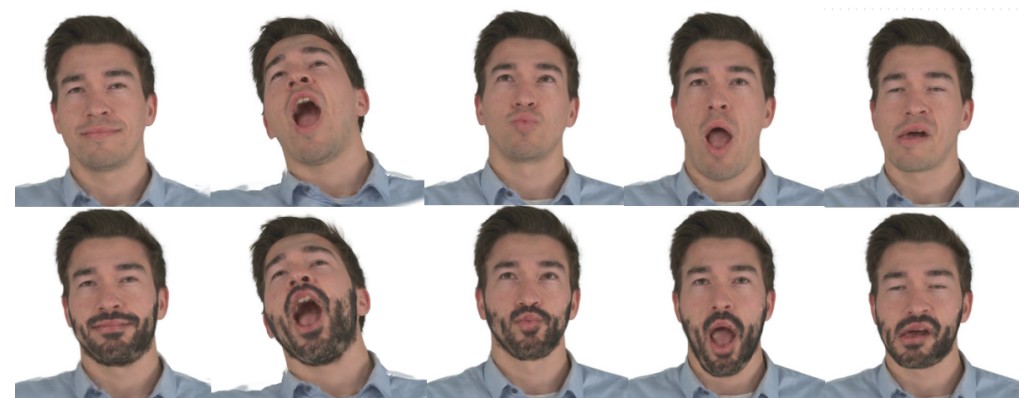

Figure 4: Dynamic behavior of face swapping. Top row: avatar 253 rendered across different poses and expressions. Bottom row: the same avatar with a beard transferred from avatar 218. The transferred segment adapts consistently to head pose and expression, following the deformations of the underlying FLAME mesh.

reason, the input coordinates $\mu_i \in \mathbb{R}^3$ are first encoded using a HashGrid (Tancik et al., 2023),

$$\mathbf{y_i} = h_\phi(\mu_\mathbf{i}) \, ,$$

with trainable encoding parameters $\phi$, producing a richer positional representation. As the parameters $\phi$ are arranged into $L$ levels, each containing feature vectors of dimension $F$, the resulting dimensionality of $\mathbf{y}$ is $\mathbb{R}^{FL}$. In our experiments, we use $L = 16$ and $F = 2$.

FaceParts model uses an additional neural network, which maps each encoded Gaussian center to the rest of the Gaussian components:

$$f_\theta \colon \mathbb{R}^{FL} \to \mathbb{R}^{52} \, ,$$

such that together with the HashGrid, both define the flow

$$g_{\phi,\theta} = f_\theta \circ h_\phi \, ,$$
$$g_{\phi,\theta}(\mu_i) = (\Sigma_i, \sigma_i, c_i) \, ,$$

where $\mu_i$ is the centroid location, $\mathbf{\Sigma}_i$ is the covariance matrix capturing anisotropic shape, $\sigma_i$ denotes opacity, and $\mathbf{c}_i$ contains the SH colour coefficients of the $i$-th Gaussian.

The core of the architecture is a low-dimensional bottleneck layer $f^{(l-1)} \colon \mathbb{R}^d \to \mathbb{R}^k$, preceding the output, where $d$ denotes the hidden layer size, set to $d = 256$ in our experiments. This bottleneck forces the network to route Gaussians with similar texture, color, or structural properties through the same channel, effectively disentangling the avatar into $k$ appearance-based groups. To avoid trivial collapse and to ensure that each Gaussian strongly activates a single channel, we apply the Gumbel-Softmax function for each latent code $z_i \in \mathbb{R}$ at the bottleneck. The Gumbel-Softmax Distribution is introduced in (Jang et al., 2016), and it provides a differentiable approximation for sampling from the categorical distribution. It is defined as

$$v_i = \frac{\exp((\log(\pi_i) + g_i)/\tau)}{\sum_{j=1}^{k} \exp((\log(\pi_j) + g_j)/\tau)} \, ,$$

where $g_1, \ldots, g_k \sim \mathrm{Gumbel}(0, 1)$, $\pi_1, \ldots \pi_k$ denote probabilities and $\tau > 0$ is a temperature parameter. The output scores $z_1, \ldots, z_k$ are interpreted as unnormalized log-probabilities, where $z_i = \log(\pi_i)$. For small enough $\tau$ the output is closer to one-hot vector. This guarantees both balanced utilization across channels and clear channel assignment, so that every Gaussian is associated with one maximally responsive bottleneck unit. After training, the reconstruction head is discarded, and Gaussians are labeled with segments $s$ according to their strongest channel activation

$$s = \arg\max_i v_i,$$

providing an initial segmentation.

Figure 5: Segmentation results for different bottleneck sizes. Smaller bottlenecks (e.g., size 3 and 4) lead to interpretable groupings such as clothing, skin, and other features, while larger bottlenecks often produce fragmented clusters without clear semantic meaning.

**Group Clustering**  The channel-based segmentation obtained in the previous step still contains mixed regions and spurious assignments. To refine this decomposition, we apply density-based clustering (DBSCAN) separately within each bottleneck group. This stage has two complementary roles: (1) filtering out noisy Gaussians that do not belong to consistent regions, and (2) subdividing groups into distinct spatially coherent clusters. Precisely, for a set of Gaussians $\mathcal{G}_s$ belonging to a chosen segment $s$, we obtain the partition

$$\mathcal{G}_s = \{\mathcal{G}_s^1, \ldots, \mathcal{G}_s^m, \mathcal{G}_s^{\mathrm{noise}}\} \,,$$

where $\mathcal{G}_s^{\mathrm{noise}}$ contains all Gaussians that were not assigned to any valid cluster and are subsequently removed. The remaining clusters $\{\mathcal{G}_s^1, \ldots, \mathcal{G}_s^m\}$ correspond to well-defined, semantically meaningful face parts such as individual facial hair regions or skin patches, which can be reliably transferred to other avatars. As our method operates on each Gaussian avatar independently, and the clustering outcome is sensitive to both the number of Gaussians and their spatial density, we empirically determine suitable hyperparameter ranges for DBSCAN. Specifically, we find that setting the minimum number of samples between 50 and 150 and the neighborhood radius $\epsilon$ between 0.001 and 0.02 yields the most stable results.

**Face Swap**  The final stage integrates extracted facial segments into a new avatar, as we demonstrate in Figure 3. As the first step, we transform the Gaussians representing the extracted segment back into the local coordinate system of the source avatar's FLAME mesh. This is achieved by applying the reverse of the operations introduced in Paragraph 4. Each Gaussian carries the index of a specific triangle to which it is bound, denoted as $b_i$. The basic strategy for merging a set of Gaussians representing a given avatar, $\mathcal{G}_X$, with an object, $\mathcal{G}_Y$, and for each mesh triangle $T$, is

$$\rho_T = \{g_i \in \mathcal{G}_X \mid b_i = T\} \cup \{g_j \in \mathcal{G}_Y \mid b_j = T\} \,,$$

where $\rho_T$ represents the combined set of Gaussians and $g = \{\boldsymbol{\mu}, \boldsymbol{\Sigma}, \sigma, \mathbf{c}\}$ represents the parameters of a single Gaussian. Since both the segment and the avatar are expressed in the local coordinate system of their respective FLAME meshes, during the merge the transferred segment is automatically aligned with the target face. To ensure smooth fusion of overlapping elements, and to avoid cases where parts of the transferred object would be occluded by avatar Gaussians, we propose two strategies: replacement and overlap. The first identifies triangles of the FLAME mesh that are likely to contain Gaussians overlapping the inserted object and removes them. The decision is based on a fixed threshold parameter $n$, which specifies the minimum number of object Gaussians required within a triangle to trigger replacement. Formally, for each triangle $T$ belonging to the object:

$$\text{if } |\rho_T^Y| > n \quad \Rightarrow \quad \rho_T = \rho_T^Y, \quad \text{else } \rho_T = \rho_T^X \cup \rho_T^Y,$$

where $\rho_T^X$ and $\rho_T^Y$ are the subsets of Gaussians from the avatar and the object, respectively.

The second strategy, overlap, applies the same principle without removing Gaussians from the avatar. For triangles where Gaussians of the object are inserted, the opacity of the corresponding Gaussians of the avatar is reduced by a factor $o \in [0, 1]$, ensuring the visibility of the transferred segment while preserving the original details. Formally, for each triangle $T$ of the object:

$$\rho_T = \{g_i(\sigma \cdot o) \mid g_i \in \rho_T^X\} \cup \rho_T^Y,$$

where $g_i(\sigma \cdot o)$ denotes an avatar Gaussian with opacity reduced by factor $o$.

Results of both strategies under different hyperparameter settings are provided in the appendix. More examples of facial parts swaps can be found in the appendix Figure 10.

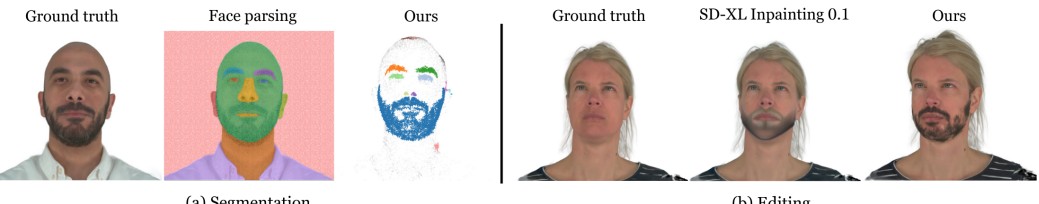

Figure 6: Example highlighting the limitations of segmentation and generative models. (a) Segmentation, left: ground-truth image. Middle: output from a 2D face segmentation model. Right: our result. (b) Editing, middle: inpainting with a generative diffusion model. Right: our result. The segmentation model struggles to capture realistic appearance and fine-grained details, while the diffusion model lacks reliability in out-of-distribution cases (e.g., woman with a beard). Our approach preserves identity and structure while producing natural and consistent facial features.

## 5 EXPERIMENTS

Evaluating our method presents a unique challenge: the quality of segmentation and face-part transfer cannot be quantified with standard generative benchmarks, as our framework does not synthesize avatars from scratch but instead recombines existing avatars and extracted segments. Moreover, several aspects of our approach, particularly the unsupervised segmentation of Gaussians into meaningful facial parts, can only be assessed visually.

For this reason, our experiments focus on two complementary objectives. The first is to verify whether the proposed disentanglement and clustering steps successfully isolate semantically coherent facial parts such as beards, mustaches, eyes, or eyebrows. The second is to evaluate whether these extracted parts can be attached to new avatars without deformation, while ensuring that the resulting face remains consistent with the underlying FLAME architecture and continues to respond correctly to its deformations.

**Dataset** We use recordings from 11 subjects in the NeRSemble dataset (Kirschstein et al., 2023), trained with Gaussian Avatars (Qian et al., 2024), which produce Gaussians spatially aligned to the FLAME mesh. The recorded sequences cover a wide range of facial dynamics, including head motions, natural expressions, emotions, and spoken language. Each Gaussian is represented by its 3D position and a feature vector of 56 dimensions: four components of a quaternion encoding rotation and three scale parameters (all normalized per point), an opacity value passed through a sigmoid activation, and spherical harmonic coefficients clipped to the interval $[-2.4, 2.4]$ to suppress outliers with negligible visual effect.

**Disentanglement and Segmentation** As the first part of our benchmark, we trained separate models for each avatar to evaluate how well the bottleneck disentangles facial and non-facial components. We varied the size of the bottleneck layer $k$ from 3 to 7, which directly controls how many groups of Gaussians are formed during segmentation. From these experiments, we observed that a bottleneck size of 3 most consistently produced meaningful and interpretable groups. Typically, one cluster corresponded to clothing, another to skin, and a third to remaining elements such as eyes, hair, eyebrows, or beard. Increasing the bottleneck size beyond 3 did not lead to finer separation of these elements. For example, facial hair and eyebrows rarely emerged as distinct groups, likely due to the strong similarity in color and texture between the Gaussians that compose them. Instead, higher bottleneck sizes often resulted in fragmented or non-semantic clusters that were difficult to interpret. An example of segmentation results across different bottleneck sizes is presented in Figure 5. The clustering process refines broad bottleneck groups into semantically meaningful facial segments. The extracted segments can be directly re-attached to other avatars.

**Face-Swapping Experiments** To assess the quality of attaching extracted facial segments to novel avatars, we conducted experiments using ten segments obtained in the previous step, spanning four

semantic classes: beard, eyebrows, eyes, and mustache. Each segment was transferred to all other avatars except its source identity, yielding eleven modified avatars per segment. In line with recent advances such as Arc2Avatar, which generates expressive 3D avatars from a single image while retaining identity via a dense correspondence mesh and blendshape-based expressions (Gerogiannis et al., 2025), we also include rigorous identity consistency and expression or pose distance metrics in our evaluation. Methods like (Liu et al., 2025) demonstrate the importance of precise region-level control and local adaptation when modifying avatars, so our segment-wise metrics (by semantic class) align with best practices in the field. Evaluation included both qualitative inspection and quantitative analysis to determine whether segment attachment affects identity preservation or the ability of avatars to follow pose and expression variations.

We report Average Expression Distance (AED), Average Pose Distance (APD), and Identity Consistency (ID), which are standard measures in prior work on editable avatars (Li et al., 2024a). Following (Deng et al., 2019), AED and APD were computed using Deep3DFace, while ID was measured using ArcFace (Deng et al., 2022). All metrics were computed pairwise between each original avatar and its modified counterparts. To capture dynamic behavior, we sampled 20 timestamps per avatar sequence spanning a range of poses and expressions, ensuring that the attached segments deform consistently with the FLAME mesh instead of remaining static artifacts. To capture dynamic behavior, we sampled 20 timestamps per avatar sequence covering a range of poses and expressions, instead of relying on a single rendered frame. This procedure ensures that attached segments deform consistently with the FLAME mesh rather than remaining static artifacts. Figure 4 illustrates this effect, where the beard adapts to pose and expression changes, confirming correct integration. Metrics were averaged across all generated avatars, and results were also grouped by semantic class to analyze the influence of different segment types. Larger regions such as beards have a stronger impact on overall scores compared to smaller parts such as eyebrows, thus per-class reporting provides clearer interpretation of performance. While a dedicated generative model is not employed in our pipeline, the absence of universal quantitative metrics for part-swapping, where evaluation is often limited to visual inspection (Wang et al., 2024a; Kim et al., 2025), motivated us to also compute Fréchet Inception Distance (FID). Here we treat avatars with transplanted segments as generated samples and compare them to their unmodified counterparts as real data. To make this comparison robust, we rendered multiple views of each avatar under varying poses, expressions, and camera positions, and used these views to calculate FID. Figure 6 presents a comparative evaluation of the standard approaches against our method. The aggregated results are summarized in Table 1.

**Hair Transplant** While our experiments primarily target small, semantically local facial parts like eyebrows, eyes, mustaches, and beards, where few prior methods operate, we also assess transferring full hair as a feasibility check. In our pipeline, hair often emerges as a distinct cluster from the disentanglement stage, which is sufficient to treat hair as a transferable segment and re-attach it to novel avatars. As hair transfer is more common in dedicated approaches such as MeGA (Wang et al., 2024a), we include a side-by-side comparison on matched source hair and target identity (Figure 7). Qualitatively, our swaps preserve style and remain responsive to pose and expression, with occasional minor artifacts where the MeGA contains visible discontinuity at the hairline. The experimental details and visualisations are provided in the appendix.

| Segment | AED ↓ | APD ↓ | ID ↑ | FID ↓ |
|---|---|---|---|---|
| Beard | 0.037 (0.015) | 0.007 (0.006) | 0.905 (0.011) | – |
| Eyebrows | 0.014 (0.006) | 0.004 (0.003) | 0.927 (0.007) | – |
| Eyes | 0.004 (0.002) | 0.001 (0.001) | 0.986 (0.003) | – |
| Moustache | 0.029 (0.014) | 0.005 (0.004) | 0.952 (0.010) | – |
| Global Avg | 0.021 (0.009) | 0.004 (0.003) | 0.943 (0.008) | 5.872 |

Table 1: Quantitative evaluation of face-part transfer. Metrics are averaged across all avatars and grouped by segment type. Results indicate that attaching an additional segment has only a small influence on expression and pose extraction. The largest variations are observed for the beard, likely because it is the largest segment. The overall FID score is also reported to quantify realism relative to unmodified avatars.

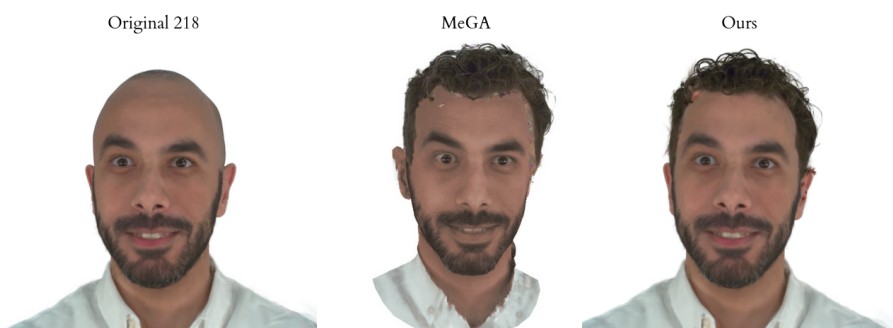

Figure 7: Comparison of transferring hair from avatar 306 to avatar 218 using our FacePart and MEGA (Wang et al., 2024a).

## 6  ABLATION STUDY

**Gumbel-Softmax** We found that employing the Gumbel-Softmax function instead of the standard softmax is crucial to better disentangle facial features. Moreover, we implemented two custom loss functions $\mathcal{L}_{usage}$ and $\mathcal{L}_{sparsity}$ (see Appendix D for details, images a-d Fig. 8 for the visual comparison), but observed that they provide only marginal improvement when combined with HashGrid. **HashGrid** A model operating directly on raw $xyz$ coordinates fails to capture fine-grained features such as eyes and eyebrows, although it can still correctly learn larger regions such as the beard or clothing (see image e in Fig. 8).

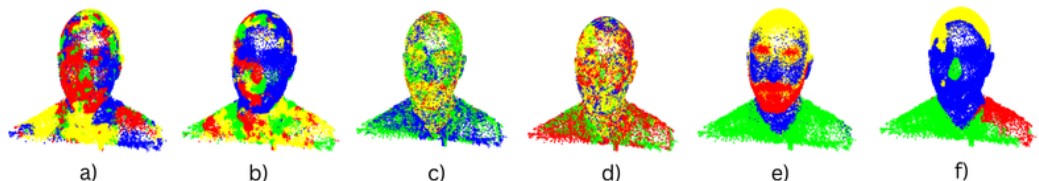

Figure 8: a) $\mathcal{L}_{usage} + \mathcal{L}_{sparsity}$ b) $\mathcal{L}_{sparsity}$ c) $\mathcal{L}_{usage}$ d) no additional loss e) direct prediction on $xyz$ coordinates f) prediction on $xyz$ with the standard softmax. The sparsity loss encourages the network to cluster nearby regions, while the usage loss promotes a more uniform bottleneck distribution across the entire avatar. Models trained without HashGrid fail to disentangle smaller facial parts.

## 7  CONCLUSION

We introduced FaceParts, a framework for unsupervised segmentation and editing of Gaussian Splatting avatars, which decomposes avatars into semantically meaningful facial parts and allows for their transfer between avatars. Our method operates directly in the Gaussian domain, enabling precise manipulation of facial features, such as beards, eyebrows, and eyes, without external supervision. By integrating feature disentanglement and density-based clustering, FaceParts provides a flexible tool for avatar customization, with minimal manual intervention.

Experiments on the NeRSemble dataset with 11 subjects demonstrate the effectiveness of FaceParts. We achieved high Identity Consistency (ID = 0.943), and lowAverage Expression Distance (AED = 0.021) and Average Pose Distance (APD = 0.004), indicating that facial parts transferred between avatars retain identity and adapt to varying poses and expressions. The FID further confirms the realism of the results. However, some limitations were noted as clustering process can produce fragmented or ambiguous facial segments when textures are highly similar.

Future work includes integrating *CLIP-based guidance* for *style transfer* to enhance facial part modifications, as well as exploring *style-conditional generative models* to further refine avatar customization. Additionally, expanding the dataset and optimizing clustering techniques will improve segmentation accuracy and robustness, especially in dynamic environments.

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

## A  REPLACEMENT STRATEGIES EXAMPLES

In this section, we present different results of combining the avatar and the segment using either overlap or replacement strategies with various hyperparameters. Examples for different parameter values while pasting a beard on the Avatar360 are shown in Figure 9.

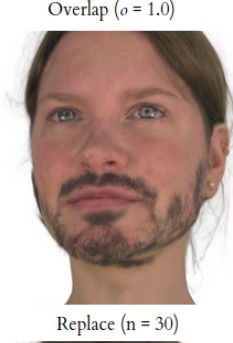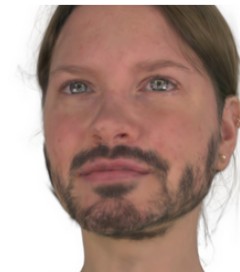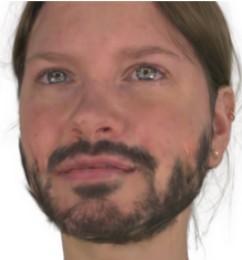
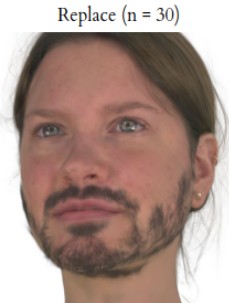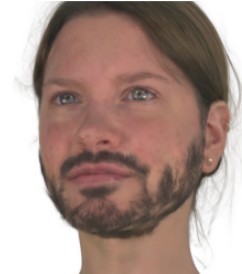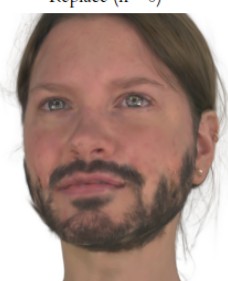

Figure 9: The first row illustrates the overlap strategy with different values of parameter $o$. Higher values make the attached beard less visible, as the avatar's Gaussians dominate. The second row shows the replacement strategy, where smaller values of parameter $n$ result in a more visible beard.

## B  FACIAL PARTS SWAPS

We used different avatars as origins from which we extracted facial parts and attached them to other avatars, modifying only a single attribute to enable clear before-and-after comparison. Effects are demonstrated in Figure 10.

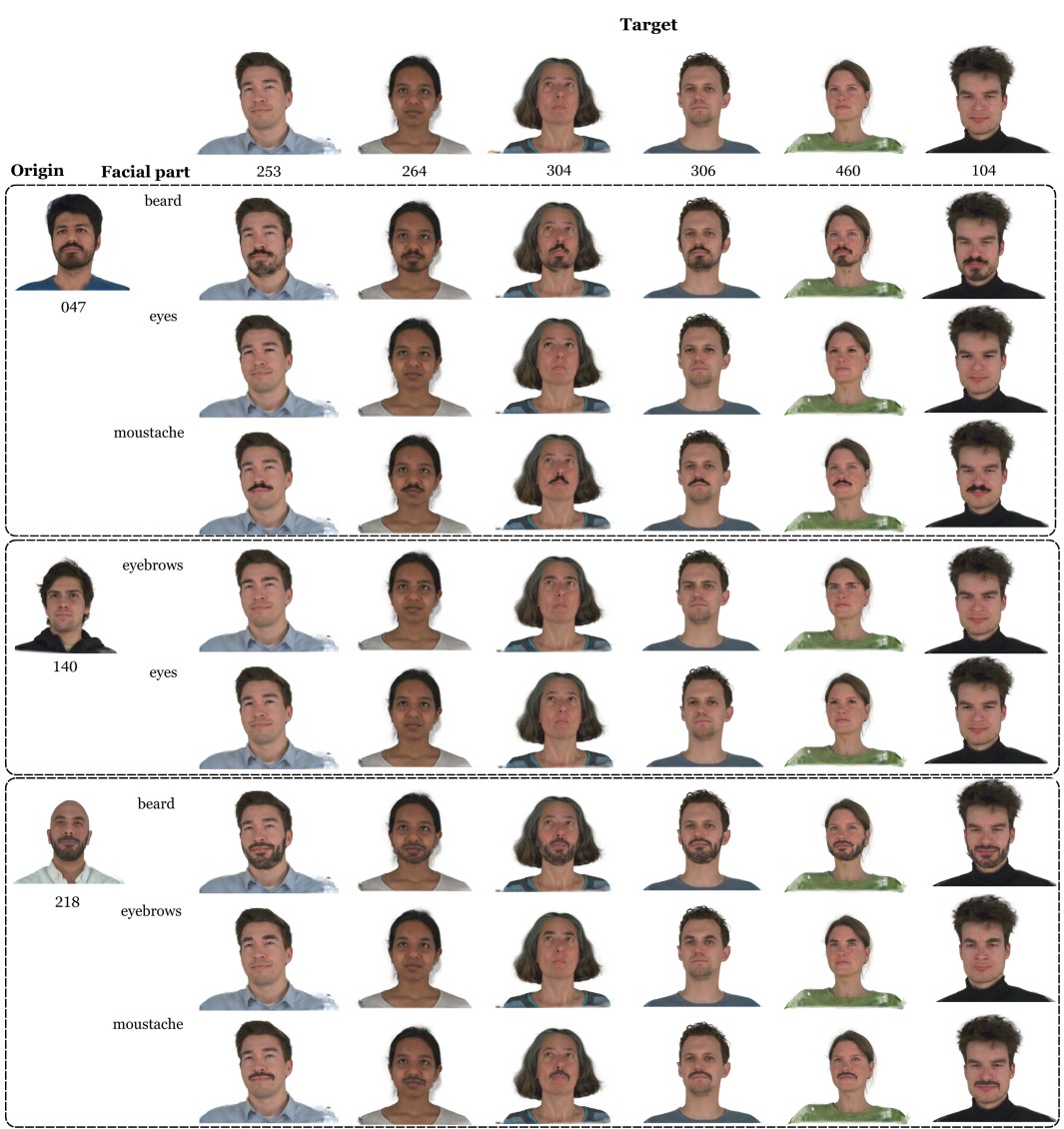

Figure 10: Examples of facial part swaps.

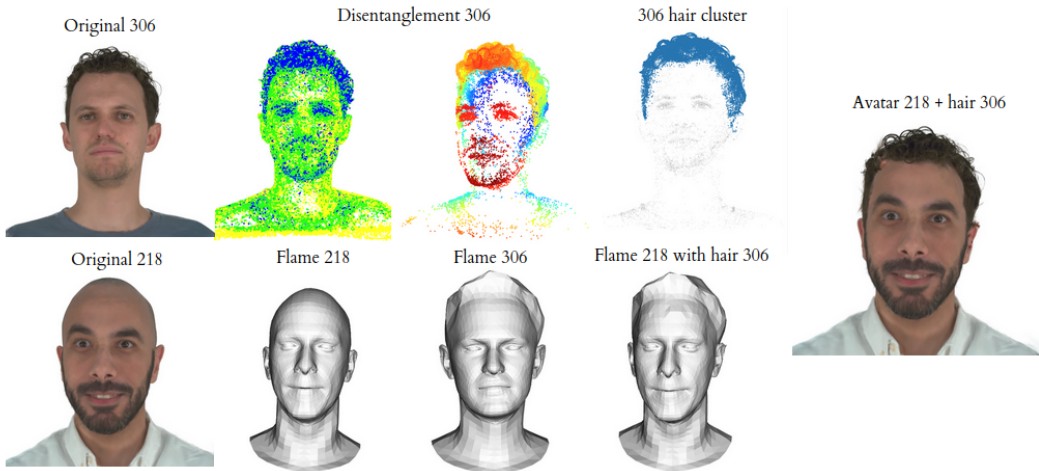

Figure 11: First row: hair extracted from avatar 306 using the disentanglement network followed by clustering. Second row: face swap with an additional step, beyond transferring Gaussians in relative coordinates, we locally adjust the FLAME mesh for avatar 218 (only the triangles corresponding to the hair region of avatar 306) for better alignment.

## C  HAIR TRANSPLANT

We prepared avatar 218 with the hairstyle from avatar 306. For comparison with MeGA, we rendered the headless avatar 218 and modified its hairstyle using the hair from avatar 306, following the configuration provided by the authors. In contrast, our FacePart method follows the standard segment transfer approach, with an additional step that transfers the FLAME parameters associated with the hair. Visualizations of the different steps of our method are shown in Figure 11

## D  LOSS FUNCTIONS

**Bottleneck sparsity loss.** This term encourages each sample's bottleneck distribution to be confident (low entropy):

$$\mathcal{L}_{\text{sparsity}} = \frac{1}{B} \sum_{i=1}^{B} \left( -\sum_{j=1}^{k} p_{ij} \log p_{ij} \right),$$

where $p_{ij}$ is the probability of bottleneck unit $j$ for sample $i$ (obtained via softmax or Gumbel-softmax), $k$ is the number of bottleneck units, and $B$ is the batch size.

**Bottleneck usage loss.** This term encourages uniform usage of bottleneck units across the batch. Let

$$u_j = \frac{1}{B} \sum_{i=1}^{B} \mathbb{1}\left[ j = \arg\max_{m} p_{im} \right]$$

be the empirical frequency of bottleneck unit $j$ being the most active. The usage distribution entropy is

$$H(u) = -\sum_{j=1}^{k} u_j \log u_j.$$

The loss is then defined as

$$\mathcal{L}_{\text{usage}} = \log k - H(u),$$

which penalizes deviation from the maximum entropy (uniform) distribution.

