# OpenReview forum: "FaceParts: Segmentation and Editing of Gaussian Splatting Avatars"
_ICLR.cc/2026/Conference — ICLR 2026 Conference Withdrawn Submission_

### Official Review · Reviewer_eQ3i · 2025-10-15

**Soundness:** 3
**Presentation:** 3
**Contribution:** 2
**Rating:** 4
**Confidence:** 4

**Summary:**

This paper introduces FaceParts, a framework for unsupervised segmentation and editing of 3D Gaussian Splatting avatars, enabling fine-grained control over facial features without any manual labeling or mesh-based supervision. The method operates directly in the Gaussian domain and decomposes avatars into semantically meaningful facial parts—such as eyes, eyebrows, beards, and mustaches—through a three-stage pipeline of feature disentanglement, density-based clustering, and FLAME-anchored part transfer. Using a Gumbel-Softmax bottleneck combined with a HashGrid encoding, FaceParts produces interpretable and spatially coherent 3D segments that can be seamlessly swapped across identities while preserving pose and expression fidelity.

**Strengths:**

The paper is easy to understand. Logic flow looks look.

The author include the code, this is great.

**Weaknesses:**

1. Limited scope of segmentation granularity.
Although the method successfully isolates macro features (beard, eyebrows, eyes), it struggles with fine-grained or texturally similar regions (e.g., distinguishing eyebrows from hair or skin patches). The paper acknowledges this but doesn’t propose a mechanism for improvement.

2. Unclear generalization beyond controlled datasets.
The experiments are conducted on 11 subjects from NeRSemble, which are high-quality, uniformly lit captures. It remains uncertain whether the method generalizes to in-the-wild or noisy Gaussian avatars (e.g., ones generated from single-view reconstruction). Since unsupervised methods often rely on texture and density patterns, robustness under domain shifts (different lighting, camera rigs) should be verified. I guess it can be zero-shot? If so, what if for other ethinities and other types of data?

3. Unclear if editing is 3D-consistent
Very often in 3D editing, the front and the side view quality diffes a lot, the authors should discuss it and analyze if this is a robust method

4. Evaluation metrics could be more task-aligned.
While ID, AED, APD, and FID are reasonable, they mostly measure global coherence rather than local edit quality. For a segmentation/editing paper, metrics such as region-overlap consistency, intra-part smoothness, or semantic coherence across views would provide stronger evidence of part-level fidelity.


5. Missing some comparisons.
The authors should also compare it with

a. "TextToon: Real-Time Text Toonify Head Avatar from Single Video", which is also an editing method based on instruct Nerf2Nerf
b. PERSE: Personalized 3D Generative Avatars from A Single Portrait

Now very small set of baselines overall, hard to evaluate

6. Limited theoretical or learning analysis.
The Gumbel-Softmax bottleneck is a neat engineering choice, but its behavior (e.g., sensitivity to τ, number of bottleneck units) is not deeply analyzed. A discussion of the latent entropy dynamics or channel utilization patterns during training would make the paper more insightful and reproducible.

7. Ambiguity in cluster merging during editing.
The two fusion strategies (replacement vs. overlap) are described qualitatively, but their trade-offs (e.g., smoothness vs. occlusion artifacts) are not quantitatively compared. The method might benefit from learned opacity blending rather than fixed heuristics.

8. Where is the demo videos or editing results
Most of 3D papers requires that, if possible, please upload them during rebuttal.

**Questions:**

Demo videos and visualization results? If the authors can add the demo videos or visual things, it would be more trustable.


For Choice of DBSCAN Parameters. How were ε and min_samples in DBSCAN determined? Do these parameters generalize across avatars with different numbers of Gaussians, or were they tuned per subject? Did the authors consider hierarchical clustering to handle multi-scale structures (e.g., separating hair from beard)?

The paper replaces raw (x, y, z) coordinates with a HashGrid positional encoding. Could the authors elaborate on why this encoding was necessary for semantic disentanglement, beyond just improving spatial detail?

Please also see the weakness.

---

### Official Review · Reviewer_w5tk · 2025-10-31

**Soundness:** 2
**Presentation:** 3
**Contribution:** 2
**Rating:** 2
**Confidence:** 4

**Summary:**

The paper addresses the task of unsupervised segmentation of 3DGS-based avatars, which enables down-stream application of transferring parts from one avatar to another, such as hair styles, eyebrows and beards.

Instead of relying on multi-view inconsistent facial segmentation masks, or attempting to optimize for a 3D segmentation which best fulfills 2D segmentation constraints, the paper proposes a creative approach to facilitate completely unsupervised segmentation.
By training a neural predictor, which is tasked to map the 3D gaussian center position to the remaining Gaussian attributes, and establishing a bottlneck with a Gumbel-Softmax function, the authors obtain self-supervised segmentations per Gaussian.
Subsequently, the segmentation masks are refined using the DBSCAN clustering algorithm.

Once such segmentations have been obtained, a single class can be isolated and transferred to a new avatar, by leveraging the Gaussian-to-traingle assignment.

The authors highlight the difficulty of evaluation such a creative and open task, but do not compare against any baselines or ablation experiments. Instead they simply evaluate FID and how well pose, expression and identity is preserved when swapping several different face parts.

**Strengths:**

- The paper follows a create and simple approach to obtain 3D/multi-view consistent segmentation, without manual steps, despite potentially tweaking some hyperparameters of the DBSCAN algorithm.
- The strategy to insert parts of one avatar to another seems to work very well, although some video results would enabled a more thorough qualitative validation.
- The of use of a Gumbel-softmax activation for unsupervised segmentation is simple and effective.
- The paper is well-written, easy-to-follow, and describes almost all technical components with a suitable level of detail.

**Weaknesses:**

- While I do acknowledge that evaluating the task at hand is difficult, the authors do not make any efforts to provide any sort of baseline or ablation experiment. As such, it is hard to put the presented quantitative numbers into perspective. For example, one could construct a rather simple baselines which e.g. takes 2D semantic segmentation predictions and lifts them to 3D by optimizing for a semantic class per Gaussian attribute.
- The user does not seem to have any sort of fine-grained control. Instead, the user is stuck with taking whatever the clusters presents, e.g. separating parts of the beard seems impossible, transferring small accessoires like earrings or piercings seems also unlikely to work.
- Overall the approach seems rather simple, but effctive. However, in combination with little evaluation and ablation it rather makes the impression of a technical report than a paper at a top tier conference.
- Although it is not 100% clear from the text, it seems that the authors manually choose hyper-parameter setting for DBSCAN per avatar.

**Questions:**

- How did the authors arrive at the task for the network $f$? Does the task for $f$ make sense (see Line 241), i.e. mapping gaussian center to remaining Gaussian attributes? It almost looks like training a neural field over gaussian parameters. What would happen if you directly cluster the gaussian attributes? instead of employing the Gumbel-bottlneck trick?
- Is there any indication that the method can go beyond what face parsing can already do? I.e. can it discover finer or novel classes in a reliable fashion?
- What happens if the clothes of a person have a similar color then the hair? or if beards and hair have the same color and no gap between them? Basically, I am a bit worried that the proposed method works well for the 11 people from GaussianAvatar, but are there any guarantees that the approach robustly works in general?
- There seems to be no before and after DBSCAN viz. of segmentation? This could further help the understanding. Similarly, I am wondering how the results in the teaser figure were obtained, i.e. how are eyebrows and eyes separated from beard class without manual intervention? Or did you manually choose a point in the beard region as starting point for DBSCAN?

---

### Official Review · Reviewer_sEc3 · 2025-11-03

**Soundness:** 3
**Presentation:** 2
**Contribution:** 3
**Rating:** 4
**Confidence:** 4

**Summary:**

This paper proposes FaceParts for unsupervised segmentation and editing of Gaussian scattering avatars. This method operates directly in the Gaussian domain, decomposing avatars into semantically coherent facial parts without supervision. It integrates feature disentanglement, density-based clustering, and FLAME-anchored part transfer, enabling accurate editing and part exchange across different avatars. Overall, the method demonstrates promising results.

**Strengths:**

* This method demonstrates reasonable unsupervised segmentation results and reasonable static transfer results.
* The bottleneck and clustering designs are reasonable and exhibit the expected effects.

**Weaknesses:**

* The experimental evaluation in this paper is not comprehensive. It lacks ablation studies and evaluation regarding the quality of self-supervised segmentation, the group clustering process, and the design of Gaussian fusion after swapping—yet these are the core contributions of the paper.
* The paper also lacks evaluation of the continuity and naturalness of the edited results, driven video visualizations should be provided.
* Contribution: "We provide a comprehensive evaluation protocol combining identity, pose, expression, and
realism metrics to assess segment transfer." is somewhat over-claimed, as the paper uses common evaluation metrics and does not employ any baseline methods. If no other baseline method is available, quantitative comparisons of ablation experiments for this method should be provided. Otherwise, these metrics alone cannot provide any information.

**Questions:**

* The transfers shown in the paper involve rigid parts attached to the head, such as beards or short hair. What would happen if the transferred part were an accessory or longer hair?
* Is it possible to provide a rough quantitative evaluation of the model’s segmentation? Perhaps by projecting it onto 2D for calculation? I understand that due to it's a unsupervised method the results may not be as good as those of 2D supervised methods, but such results would be very helpful for understanding the quality of the self-supervised segmentation.
* The paper does not show any dynamic results (qualitative/video quantitative), so it is impossible to know whether the transfer would cause problems in expression/head pose driven animation, which makes it difficult to have confidence in the performance of the method on dynamic driving sequences.
* I noticed that in Figure 3, the beard of avatar 218 is segmented well. However, after being applied to avatar 253 in Figure 4, there is a partial loss of the beard on the right cheek of the avatar. Is this caused by the replacement and overlap strategies during the fusion stage? Could you provide a more detailed ablation study on this part? For example, what would happen if you applied only overlap or as much replacement as possible?
* Just curious, why did you choose 56 as the size of the implicit feature vector?

---

### Official Review · Reviewer_S9Lk · 2025-11-05

**Soundness:** 3
**Presentation:** 2
**Contribution:** 2
**Rating:** 2
**Confidence:** 5

**Summary:**

This paper aims to achieve unsupervised segmentation of 3D Gaussian head representations, dividing them into semantically meaningful regions (e.g., skin, hair, eyebrows) and leveraging these segments for face swapping. The main contribution lies in the unsupervised segmentation framework, which introduces an auxiliary network that produces several bottleneck feature channels as coarse clusters before predicting the final Gaussian parameters. The authors apply Gumbel-Softmax on the bottleneck features to encourage discrete part assignments. After the reconstruction stage, the latter branch is detached, and density-based clustering is applied to obtain cleaner, more consistent part groupings.

**Pros**

+ The paper presents an interesting and practical editing scenario (face swapping) built upon 3D Gaussian head representations.

+ The proposed unsupervised segmentation pipeline is intuitive, clean, and provides good segmentation on fur-related attributes.

+ The writing is clear and easy to follow, making the technical ideas accessible.

**Cons**

Although the work is more engineering-oriented and less technically novel, this is not my main concern, as conceptual simplicity is not necessarily a drawback. However, my main concerns lie in the insufficient experimental validation and the universality of the proposed pipeline, as detailed below.

- Very Insufficient Experiments. 1) There are no quantitative comparisons with existing methods, either for the segmentation task or for the face editing task. The only comparisons are qualitative examples shown in Fig. 6 and Fig. 7, each with just a single case, which could be cherry-picked and thus insufficient to support the claimed performance. Moreover, Table 1 claims “high performance” without providing any comparative baselines or reference methods, making the claim unconvincing. 2) The paper involves many engineering design choices, but no quantitative ablation studies are included. As a result, it is difficult to assess which components actually contribute to the results, where the method’s limitations lie, or what concrete insights can be drawn from the experiments.
- Limited Universality. Both the segmentation and swapping modules raise concerns about generality. 1) The segmentation results appear to work mainly for fur-related regions (e.g., hair, beard), but it is unclear how the method handles facial skin, makeup, or accessories such as glasses. This concern is raised by the visualization of Fig. 5, which shows that the facial skin region is segmented in a very fragmented and inconsistent manner 2) The granularity of segmentation is uncontrollable. For instance, in Teaser, the “beard” region incorrectly includes the nostrils, indicating unstable clustering boundaries. 3) The face swapping mechanism relies on a strong assumption that each FLAME vertex corresponds one-to-one to a Gaussian primitive, and that the source and target faces share similar geometry and facial feature distributions. When this assumption fails, I can imagine noticeable misalignment artifacts will occur, which also can be seen for the visualization in the paper and provided github page (e.g., the rightmost image on the GitHub page and Fig. 10 (pair 218→104) show mis-swapped beard regions) These issues should be discussed and analyzed experimentally, as they directly affect the robustness and universality of the approach. The paper should also include novel-view experiments to validate how well the swapped regions maintain geometric and visual consistency in 3D, as current results are limited to single-view visualization.

- Other minor weaknesses. The hairstyle swapping procedure is underspecified. In the supplement (L724–L730), the authors state that they first deform the target (ID 218) FLAME mesh to fit the source (ID 306) hairstyle before editing. It is unclear how this deformation is optimized: What is the objective? What supervision or constraints prevent identity drift of the target face during the deformation?

Overall, the above weaknesses are substantial and would be difficult to address within a short revision cycle. Therefore, I would recommend rejection for the current version. However, I am open to revising my score if the authors can provide stronger experimental evidence, clearer analysis of assumptions, and additional validations during rebuttal period.

**Strengths:**

See above

**Weaknesses:**

See above

**Questions:**

See above.

---

### Note · Authors · 2025-11-18

**Comment:**

We will correct paper according to rewiever directions.

**Withdrawal Confirmation:**

I have read and agree with the venue's withdrawal policy on behalf of myself and my co-authors.